# A Study on Physical and Rheological Properties of Rubberized Bitumen Modified by Different Methods

**DOI:** 10.3390/ma12213538

**Published:** 2019-10-29

**Authors:** Ben Zhang, Huaxin Chen, Honggang Zhang, Dongliang Kuang, Jiayu Wu, Xiaoliang Zhang

**Affiliations:** 1School of Materials Science and Engineering, Chang’an University, Xi’an 710064, China; 2015031003@chd.edu.cn (B.Z.); wjy1991@chd.edu.cn (J.W.);; 2Guangxi Transportation Research & Consulting Co., Ltd., Nanning 530000, China; zhang_paper0@163.com

**Keywords:** rubberized bitumen, modification methods, TOR, sasobit, microwave, physical and rheological properties

## Abstract

Crumb rubber (CR) manufactured from waste tires used in bitumen could improve bitumen performance and reduce environmental pollution. In this work, three different modification methods, pretreatment of the CR particles (microwave activation), warm mix additive (Sasobit), and trans-polyoctenamer (TOR) were used to improve the compatibility of CR with bitumen. Moreover, two other specimens, Sasobit and microwave activated and TOR and microwave activation were fabricated, and their performance was investigated. The softening point, elastic recovery, force ductility, rotational viscosity, temperature sweep, frequency sweep, and multiple stress creep and recovery (MSCR) tests were measured to evaluate the physical and rheological properties of rubberized bitumen. The results showed that TOR improved the physical properties of rubberized bitumen significantly but had a negative effect on the viscosity. Sasobit decreased the viscosity of rubberized bitumen considerably and improved the physical properties of rubberized bitumen moderately. Microwave treatment on CR had a negative effect on the high temperature performance and elastic recoverability of rubberized bitumen, however, attributing to the degradation and devulcanization effect of microwave on CR, the viscosity of rubberized bitumen was improved. From the results of composite modification, the influence of TOR on the performance of rubberized bitumen was more obvious than that of the microwave activation treatment. Moreover, the influence of Sasobit on its performance was less than that of the microwave activation treatment.

## 1. Introduction

Rubberized bitumen has been applied widely in bitumen pavements because of its remarkable performance, eco-friendly characteristics, and energy-saving features, etc. [1,2]. Adding crumb rubber (CR) to bitumen can effectively reduce high temperature permanent deformation, low temperature thermal cracking, load-associated fatigue cracking, and chemical aging or hardening of bitumen pavements [3,4]. Currently, two technical routes have been adopted as the primary approaches to produce rubberized bitumen which are wet and dry processes, with the wet process being used most widely today. For the dry process, CR is regarded as particles distributed directly in the aggregate of bitumen mixture and there is limited interaction between CR and bitumen, whereas, during the wet process, CR is mixed with bitumen at a high temperature by a thermomechanical process and a complete interaction exists between CR and bitumen. It can significantly improve engineering performance as compared with conventional paving grade bitumen, however, the high temperature storage instability and high viscosity of rubberized bitumen have restricted its application [5,6,7]. The different density and polarity between CR particles and bitumen leads to incompatibility and sedimentation in rubberized bitumen [8]. Furthermore, the amount of aromatic oil absorption and CR particle swelling increases the viscosity of rubberized bitumen [9]. In order to solve these problems, researchers have proposed different methods to enhance the compatibility and interaction between CR and bitumen [10,11,12]. Generally, these methods can be grouped into the following three categories: reduce the viscosity by adding warm mix additives (WMA) or blend oil additives, pretreat CR particles with microwave irradiation or chemical reagents, and improve CR compatibility with bitumen by adding preswelling agents or reactive additives.

Modifying the chemical bonds between the functional groups of the modifying agents and bitumen compounds and promoting swelling reactions are useful ways to mitigate phase separation between CR and bitumen [4,5,6,7,11]. Polyphosphoric acid, crosslinking agent, nanosilica, etc., have been incorporated in rubberized bitumen to improve the compatibility of CR particles with bitumen, among which trans-polyoctenamer (TOR) has shown high reactivity [13,14,15,16,17]. TOR is a type of polymer that consists of linear and cyclic macromolecules that can facilitate the interaction between CR and bitumen via crosslinking reactions with double bonds and the sulfur component in the bitumen and surface of the CR particles. It has been noted that the addition of TOR improved the elasticity, modulus, and temperature susceptibility of rubberized bitumen [4]. Liu indicated that there was a complex chemical reaction between TOR and rubberized bitumen that changed the rheological properties of rubberized bitumen and contributed to the bitumen binder’s storage stability [11]. In addition, TOR added to rubberized bitumen mixture can improve the high temperature resistance to rutting and low temperature crack resistance of bitumen pavements [18,19,20].

High viscosity caused by aromatic oil absorption and the CR particle swelling is another problem that needs to be solved. Many researchers have been evaluating the performance and investigating the mechanism of warm mix rubberized bitumen, which can decrease viscosity significantly and conserve fossil fuel [10,12,21,22,23,24,25,26,27,28,29]. On the basis of the chemical composition and modification mechanism, the various WMAs can be grouped into three categories: chemical, organic, and foaming additives or technologies [28]. Researchers have analyzed the workability, mechanical, fatigue, and rheological property characterization of bitumen mixtures modified with various WMAs, and found that the viscosity of rubberized bitumen reduced significantly [21,22], however, WMAs had adverse effects on the fatigue property and rutting resistance, with the exception of Sasobit additive. Xiao et al. indicated that the binder type played a key role in determining the rheological properties of WMA binders by using Fourier transform infrared (FTIR) analysis [30]. Furthermore, the modification mechanisms of WMAs were explored by researchers using a set of advanced testing equipment [28]. On the basis of a series of chemical analyses, Yu et al. found that no major chemical reactions were detected after the addition of CR and WMA, but a better dissolution of CR was achieved in rubberized bitumen with Sasobit [27]. However, some studies indicated that the stiffness of rubberized bitumen increased at low temperatures when the WMA was added, which led to cracking of the bitumen pavements [27,31,32].

The desulfurization and degradation behavior of CR particles in bitumen has an important influence on the performance of rubberized bitumen. Several studies have indicated that short-term microwave irradiation improved the devulcanization of CR and reduced its degradation [7,33,34,35]. Yu et al. found that microwave irradiation cleft the surface vulcanization network of CR, resulting in higher surface activity, and improved its compatibility with bitumen [34]. According to the microstructures of CR treated by microwave irradiation, the CR surface became fluffy, which resulted in good affinity, permeability, and reactivity at the interface with oil [36]. Liang indicated that microwave irradiation activated CR and increased the risk of fatigue cracking caused by the hardening effect of aging but endowed the aged residue with stronger resistance to thermal cracking. These results are attributed to the effect of degradation and devulcanization of CR [7], however, CR treated by microwave irradiation had an adverse effect on the stability of rubberized bitumen.

As the most commonly applied method of rubberized bitumen preparation, few studies focused on the comparison of the influence of microwave activation, Sasobit, and TOR on rubberized asphalt. Therefore, different modifications of rubberized bitumen were prepared under the same process in this study. The effect of each mothed on the physical and rheological properties of rubberized bitumen was evaluated and discussed. Moreover, two other specimens, Sasobit/microwave activated and TOR/microwave activation were fabricated and their performance was investigated. 

## 2. Experimental

### 2.1. Materials

The basic properties of SK-90 # bitumen were tested according to JTG E20 [37] and the results are illustrated in Table 1. CR was produced from vulcanized tire rubber using the ambient grinding process. CR with a 40-mesh size was used and the content was 24% by weight of the base binder.

The microwave activated rubber was processed in an ultrasound microwave (Midea Co. Ltd, Guangdong, China) oven at 800 W power for 150 s. CR was be dried and dehydrated in a thermostatic oven at 60 °C to ensure the moisture is less than 0.1%. The weight of CR was controlled at 100 g each time to ensure uniform and sufficient irradiation. The content of microwave activated rubber was 24% by weight of the base binder. Figure 1 and Figure 2 show the microstructure of CR and CR treated by microwave irradiation. It can be seen that the surface of the CR particles is relatively flat and smooth, however, the surface of those treated by microwave irradiation is porous and uneven. This porous and uneven surface can increase the reaction area between the bitumen and CR, and therefore promote the CR swelling. TOR additive is a solid polymer with a double bond structure that can crosslink the sulfur of the bitumenene and the sulfur on the rubber powder’s surface. The TOR content recommended is 4.5% by weight of CR [11]. The wet process method was used in this study to mix the binder and CR, in which the two were mixed with the base bitumen before introducing the bitumen into the aggregate.

The WMA used in this research was Sasobit additive, which is a powder and opaque pellet polymer. The content of Sasobit was 1.0% by weight of bitumen [30]. The method used to add the Sasobit additive was the same as that for TOR, in which the CR particles and Sasobit were mixed with the base bitumen simultaneously.

### 2.2. Samples Preparation

High temperature and high shear can promote the desulfurization and depolymerization of CR in bitumen significantly, which improves the binder’s elasticity [38]. It has been proven that dissolution of the CR particles is marginal at 160 °C and considerable above 200 °C [17]. The preparation of rubberized bitumen, in this research, adopted the melt blending method under high temperature and intensive shearing using a high shear mixer. The main steps are given as follows: (1)The base binder was heated to 150 °C. Then add a certain amount of the CR particles and the additive stirring at 300 rpm for 30 min;(2)Shear the CR particles and bitumen blend at 4000 rpm for 60 min at 200 °C;(3)Stir the CR particles and bitumen blend at 300 rpm for 60 min;(4)Swell the prepared rubberized bitumen in an oven for 30 min at 140 °C.

Six rubberized bitumen samples were prepared in this study, which were original rubberized bitumen (RB), TOR modified rubberized bitumen (TRB), Sasobit modified rubberized bitumen (SRB), microwave activated rubberized bitumen (MRB), Sasobit/microwave activated composite modified rubberized bitumen (SMRB), and TOR/microwave activation composite modified rubberized bitumen (TMRB).

### 2.3. Analysis Methods

This research adopted the softening point test, elastic recovery test, and force ductility test (FDT) to evaluate the physical properties of rubberized bitumen. In addition, the specimen used for the elastic recovery test was chosen for FDT to produce a test specimen with a constant cross section to reduce the error caused by the uneven distribution of the CR particles.

The Brookfield rotational viscometer and dynamic shear rheometer (DSR) was used to investigate the high temperature rheological properties of rubberized bitumen. The viscosity was tested at four different temperatures (135, 150, 165, and 180 °C). The complex shear modulus (G*) and phase angle (σ) values were measured by the temperature sweep test and frequency sweep test. In addition, the multiple stress creep and recovery (MSCR) test was performed on rubberized bitumen to assess its recoverability after unloading. It was conducted over 10 cycles of 1 s loading (creep) followed by 9 s rest (recovery) at stress levels of 0.1 and 3.2 kPa, which represents a high stress level on the pavement. *J*_nr_ and percent recovery (*P*) can be calculated using Equations (1) and (2). More details of these tests and the normative followed are shown in Table 2.

(1)$$J_{\mathrm{nr}}\;=\;\frac{\mathrm{Unrecoveraed}\;\mathrm{shear}\;\mathrm{strain}}{\mathrm{Applied}\;\mathrm{shear}\;\mathrm{stress}}$$

(2)$$P\;=\;\frac{Recoverd\;strain}{Total\;strain}\;\times \;100$$

## 3. Results and Discussion

### 3.1. Physical Properties of Rubberized Bitumen

#### 3.1.1. Softening Point

The softening point is representative of the typical properties and heat resistance of conventional bitumen, and can be used as an index of the interaction between the CR particles and bitumen [39]. As shown in Figure 3, the softening point of rubberized bitumen increased obviously as compared with base bitumen, which is attributable to the addition of CR. It has been reported that when CR interacted with bitumen, the CR particles absorbed the aromatic oil, which led to an increase in the resin and bitumenene contents in the bitumen and the volume of CR. This improved the high temperature properties of modified bitumen significantly [40,41]. The softening point of MRB decreased by 3 °C as compared with that of RB, as the microwave treatment degraded the CR particles in part. The softening points of TRB and TMRB increased significantly as compared with those of RB and MRB, which indicated that TOR was beneficial in increasing the high temperature performance of rubberized bitumen. The distribution of CR in the bitumen became more uniform because of TOR, which affected the performance of rubberized bitumen directly [11]. The softening point of SRB improved slightly as compared with that of RB, while the softening point of SMRB increased by 2.8 °C as compared with that of MRB, which indicated that Sasobit had a more pronounced effect on the softening point of rubberized bitumen contained CR with a higher degree of degradation.

#### 3.1.2. Elastic Recovery

High elastic recoverability is a key difference between rubberized bitumen and other modified bitumen. It is believed that elastic recovery can reduce bitumen pavement’s high temperature rutting and low temperature cracking [41]. Figure 4 presents the effects of modification methods on elastic recovery of rubberized bitumen. As the figure shows, incorporating CR into base bitumen increased its elastic recovery dramatically. Compared with RB, the elastic recoverability of other rubberized bitumen decreased to varying degrees, except for TRB. Virgin binder barely exhibited recovery at 25 °C, and conversely, was viscous at an intermediate temperature [17]. Therefore, the elastic recoverability of rubberized bitumen depends primarily on the elasticity of CR. This is the reason that MRB exhibited minimal elastic recovery. Swelling can stimulate rubberized bitumen to form an elastic structure, and incorporating TOR increased the elastic recovery of rubberized bitumen significantly. Sasobit had a limited influence on the elastic recovery of rubberized bitumen, which may be attributable to its limited interaction with CR.

#### 3.1.3. Force Ductility

It has been reported in the literature that FDT can reflect the ductility of bitumen in the stretching process, and evaluate on the cohesive strength of polymer modified bitumen effectively [42]. Figure 5 shows a typical rubberized bitumen force ductility curve in this research. Maximum load (*F_max_*), ductility, and fracture energy (the integral of force and ductility) were achieved to evaluate the low temperature performance of rubberized bitumen. From Figure 5, it can be seen that there is a linear relationship between the force and elongation until the peak of the force was attained *F_max_*. Some researchers have claimed that the force was influenced by the performance of bitumen in this region [42,43]. Because of the flow of the bitumen, force declines after the reaching *F_max_*. Thereafter, there was a slight increase in resistance attributable to the elasticity recovery of the CR particles, which prevents the viscous and extensional flow of bitumen [44,45]. Eventually, the specimen reached failure ductility, which results in a second peak of the force.

In Figure 6 and Figure 7, it can be seen that the ductility of RB still reached 225 mm, which indicated that there has been a certain chemical modification effect on rubberized bitumen. The chemical modification of rubberized bitumen is attributable largely to the desulfurization and depolymerization of CR under the action of high temperature and shearing, and the small molecules of the particles degrade integrate into the bitumen and change its composition. Therefore, the ductility of MRB was larger than that of RB by 26 mm and its *F_max_* was less than that of RB. The microwave irradiation broke down parts of the chemical bonds in CR, not only on its surface, but internally as well. The depolymerized molecule products can improve the flexibility of rubberized bitumen, so its ductility was greater than that of others, and it exhibited the minimum *F_max_*. The ductility of TRB and TMRB, which decreased by 18.5 mm and 15 mm, respectively, was lower than that of RB, however, *F_max_* of TRB was much larger than that of RB. The incorporation of TOR had no effect on the desulfurization and depolymerization within CR but acted on their surface with the binders. Furthermore, TOR improved the swelling of the CR particles effectively. The lighter bitumen components have a significant influence on the flexibility of rubberized bitumen and expanded when they penetrated into the CR particles [38]. Therefore, the ductility of TRB was lower than that of RB. The ductility of TMRB also was poorer than that of RB, which seems to indicate that TOR had a more significant effect on the flexibility than microwave activation did. The ductility of SRB and SMRB were close to that of RB, and the differences were 5.5 mm and 12 mm, respectively. This indicated that Sasobit had little influence on the flexibility of rubberized bitumen, and its addition to rubberized bitumen primarily played a role in physical blending. From Figure 8, it can be seen that the cohesive strength of modified rubberized bitumen was larger than that of RB.

### 3.2. Rheological Properties of Rubberized Bitumen

#### 3.2.1. Viscosity

As shown in Figure 9, when the temperature increased, the viscosity of rubberized bitumen decreasing. The viscosity of rubberized bitumen increased markedly as compared with the base bitumen at 135 °C. This can be explained by the amount of aromatic oil absorption and swelling of the CR particles [9]. Some researchers have indicated that the n-alkanes and n-alkylbenzenes have the highest propensity to penetrate into the CR particles [38]. The viscosities of TRB and TMRB were greater than that of RB because TOR increased the swelling effect in bitumen effectively [46]. The volume of CR particles increased with full swelling, and the light component of bitumen entered into the CR particles, which changed the composition of bitumen. The viscosities of SRB and SMRB were less than that of RB, because of the light component Sasobit introduced. With respect to the rubberized binders that contained Sasobit, it has been reported that they form a homogeneous solution with the base binder when stirred and cause a reduction in the viscosity of binder [18]. The viscosity of MRB was lower than that of RB, which was caused by the destruction of the crosslinking within the CR particles. The high-temperature shear in the preparation process would intensify the degradation and desulfurization of CR further, and lead to the destruction of the CR.

#### 3.2.2. Temperature Sweep Tests

Figure 10 shows that as the temperature increased, the phase angle of modified rubberized bitumen, first, decreased slightly and, then, increased. The phase angle was lowest at 50 °C. When the temperature exceeded 55 °C, it increased as the temperature increased. When the temperature reached 80 °C, the maximum phase angle of rubberized bitumen was less than 65°. Previous research has shown that when the phase angle of base bitumen was greater than 75°, the complex modulus, G*, and loss modulus, G", were substantially the same, at which point the elastomeric composition of binder was essentially negligible. Therefore, 75° can be considered the upper limiting temperature at which the binder exhibits elastic characteristics [47]. The rubberized bitumen still demonstrated prominent elasticity at 80 °C, which indicated that the presence of the CR particles had the most significant influence on the phase angle and played a vital role in viscoelasticity. The smooth or undulating change in the phase angle over a large temperature range largely was attributable to the addition of modifiers [40]. This phenomenon improves the resistance of bitumen pavements to permanent deformation [48].

The phase angles of TRB and TMRB were smaller than those of other modified rubberized bitumen. TOR improved the swelling of CR in the bitumen, which increased its elasticity obviously. The phase angle of MRB was the largest, indicating that its viscous component was larger than that of other modified rubberized bitumen. This result indicated that the degree of CR degradation had an important influence on the viscoelasticity of rubberized bitumen. The phase angle of SRB was lower than that of RB. It can be seen that the effect of Sasobit modifier on the viscoelasticity was less than that of the microwave activated CR particles. The values of G^⁎^ decreased as the temperature increased and had almost parallel upward curves that showed the linear dependence of log G^⁎^ with temperature. As expected, G^⁎^ of TRB had the highest value and was much greater than that of the remaining samples.

#### 3.2.3. Frequency Sweep Tests

It can be seen from Figure 11 that the phase angle of rubberized bitumen decreased as the frequency increased. The phase angle maintained a small change at 60 °C, which could be attributed to the addition of the CR particles [49]. The phase angles of TRB and TMRB were smaller than those of the others, as was their viscosity. This was largely because TOR improved the CR swelling in the bitumen. The phase angle of MRB was the largest, indicating that its viscous component was larger than that of other modified rubberized bitumen. SRB had a smaller phase angle than RB did, and this indicated that the effect of Sasobit was less than that of microwave activated CR particles on the viscoelasticity.

#### 3.2.4. Multiple Stress Creep Recovery

Figure 12 shows the results of MSCR tests at stress levels of 0.1 kPa in the first 100 s and 3.2 kPa in the second 100 s. The *P* values confirm that large cumulative strain applied to base bitumen showed an almost insignificant strain recovery during unloading [42]. According to the Figure 12, the sawtooth-shaped curve indicates that rubberized bitumen was characteristic of highly elastic binders. This is primarily because the presence of elastic CR particles causes changes in the recovery cycle that lead to delayed elastic strain recoveries. Table 3 shows the results of *J*_nr_ and *P* of rubberized bitumen, which are useful to characterize the delayed viscoelastic response. As the table shows, TRB had the highest percent recovery, indicating that it recovered a greater portion of deformations after unloading. Although *P* values of MRB decreased as compared with that of RB, it still maintained higher recoverability. Moreover, the results of the *P* values reported in the table were consistent with those deduced from the linear viscoelastic properties.

The *J*_nr_ value is an another MSCR parameter that indicates the rutting resistance of bitumen binder specimen. This parameter represents the potential for accumulated permanent deformation as a result of loading and unloading cycles. The lower the *J*_nr_ value, the more the bitumen binder resists rutting. The *J*_nr_ values of various rubberized bitumen show the opposite trend to the *P* value results. This provided further proof of the influence of various methods on the anti-rutting performance of rubberized bitumen.

## 4. Conclusions

In this study, the effects of TOR, Sasobit, and microwave treatment on the physical and rheological properties of rubberized bitumen were evaluated according to dynamic viscoelastic and flow behavior. Some preliminary conclusions that can be drawn from the research are the following:TOR significantly improved the high temperature stability and elastic recoverability of rubberized bitumen as compared with the other modification methods, however, as a result of the CR particles and bitumen bonds and the promoted swelling reaction caused by TOR, the viscosity of rubberized bitumen increased considerably;The viscosity of rubberized bitumen decreased significantly with the addition of Sasobit, but the effect of Sasobit on rubberized bitumen viscosity is inferior to the microwave-activated methods. Sasobit had a certain improvement in the physical and rheological properties of rubberized bitumen, but this improvement was less significantly than TOR additives;Microwave activated rubberized bitumen had the lowest viscosity, but it had an adverse effect on the high temperature performance and elastic recoverability of rubberized bitumen;From the results of composite modification, the influence of TOR on the performance of rubberized bitumen was more obvious than that of the microwave activation treatment. Moreover, the influence of Sasobit on its performance was less than that of the microwave activation treatment.

## Figures and Tables

**Figure 1 materials-12-03538-f001:**
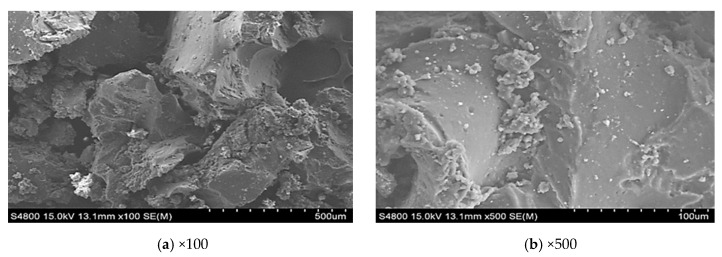
SEM image of the CR particles.

**Figure 2 materials-12-03538-f002:**
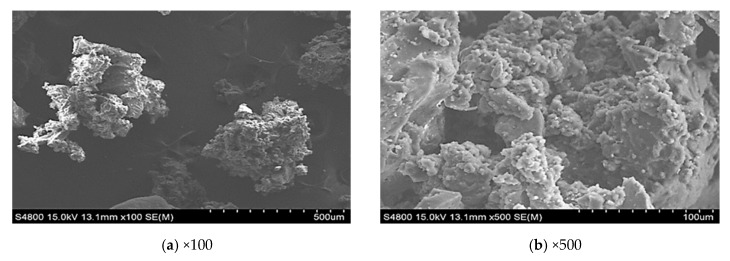
SEM image of the microwave activated crumb rubber (CR) particles.

**Figure 3 materials-12-03538-f003:**
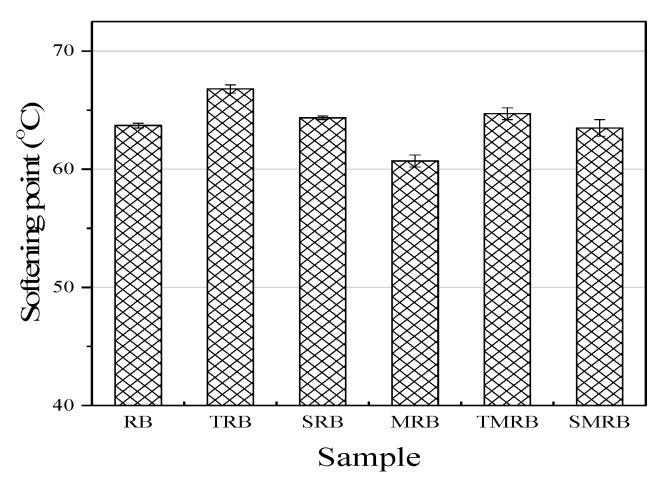
Softening point of rubberized bitumen.

**Figure 4 materials-12-03538-f004:**
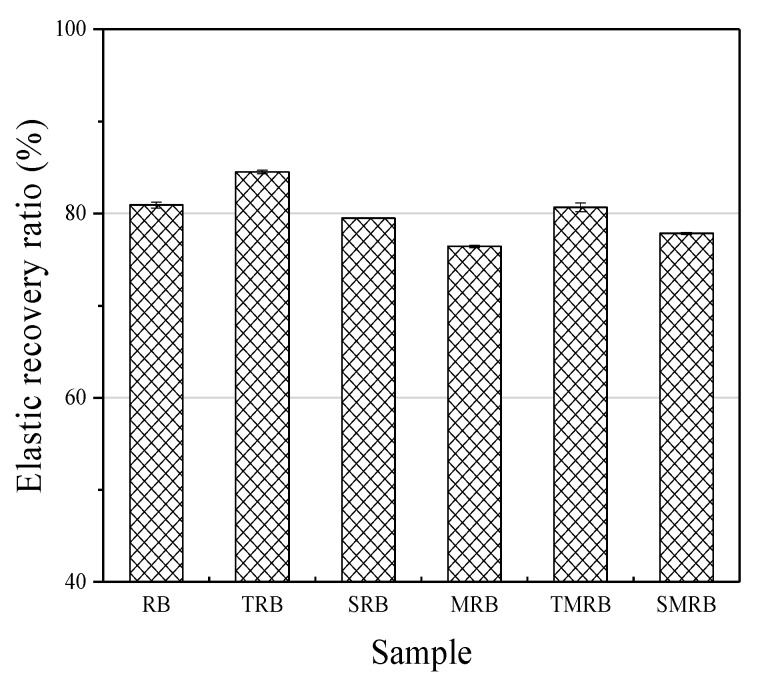
Elastic recovery of rubberized bitumen.

**Figure 5 materials-12-03538-f005:**
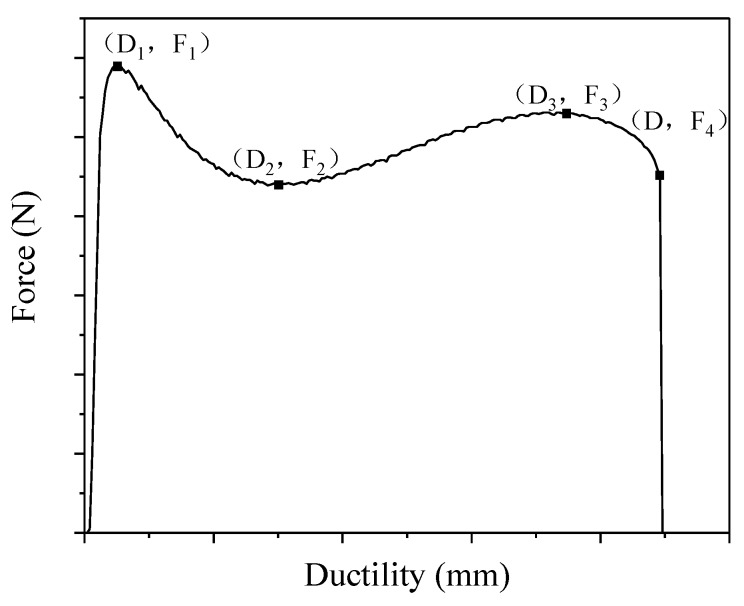
Typical rubberized bitumen force ductility curves.

**Figure 6 materials-12-03538-f006:**
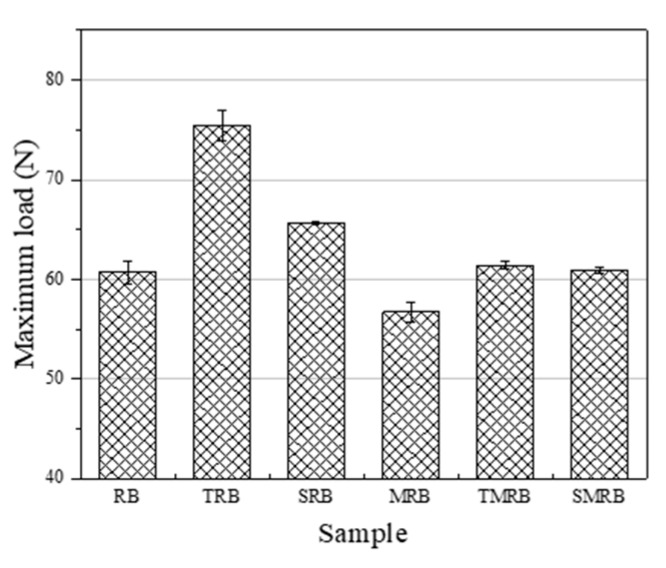
Maximum load of rubberized bitumen.

**Figure 7 materials-12-03538-f007:**
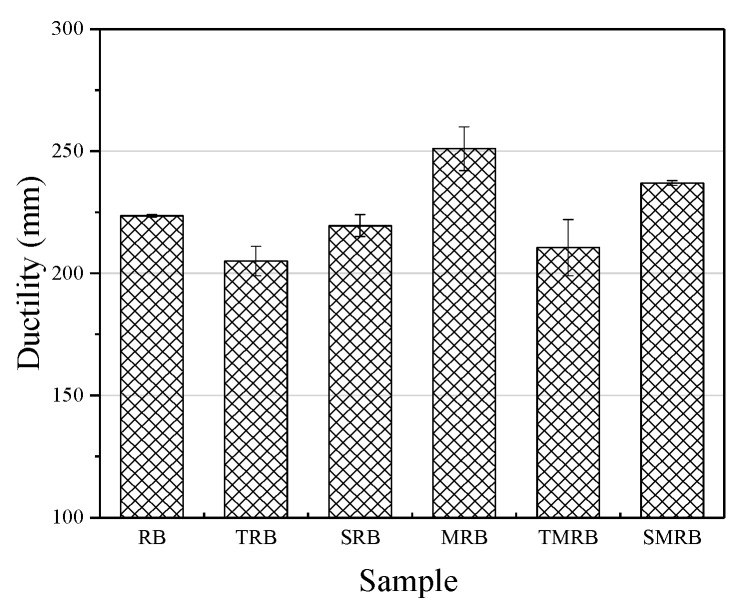
Ductility of rubberized bitumen.

**Figure 8 materials-12-03538-f008:**
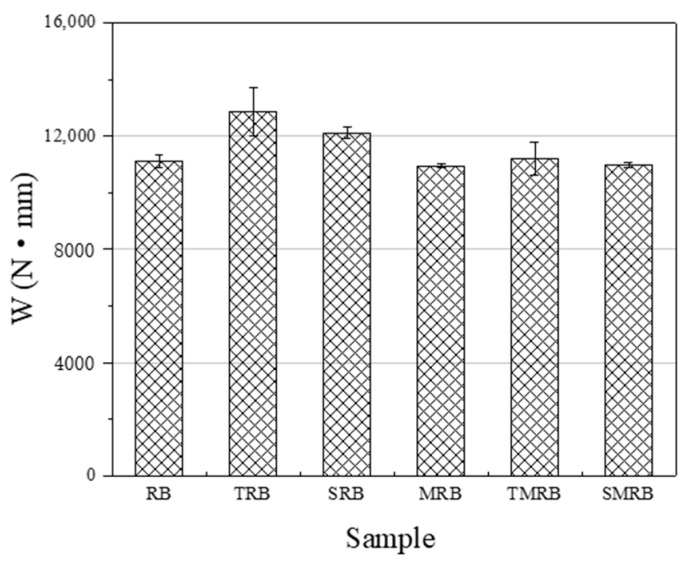
Fracture energy of rubberized bitumen.

**Figure 9 materials-12-03538-f009:**
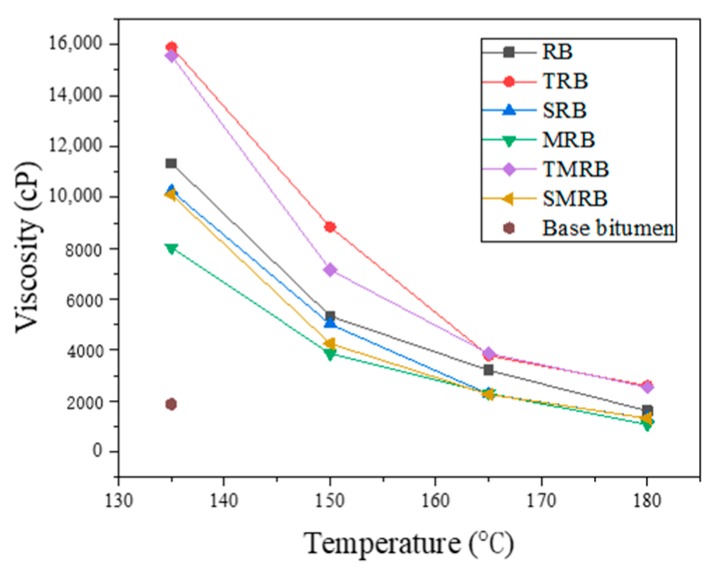
Viscosity of rubberized bitumen at different temperatures.

**Figure 10 materials-12-03538-f010:**
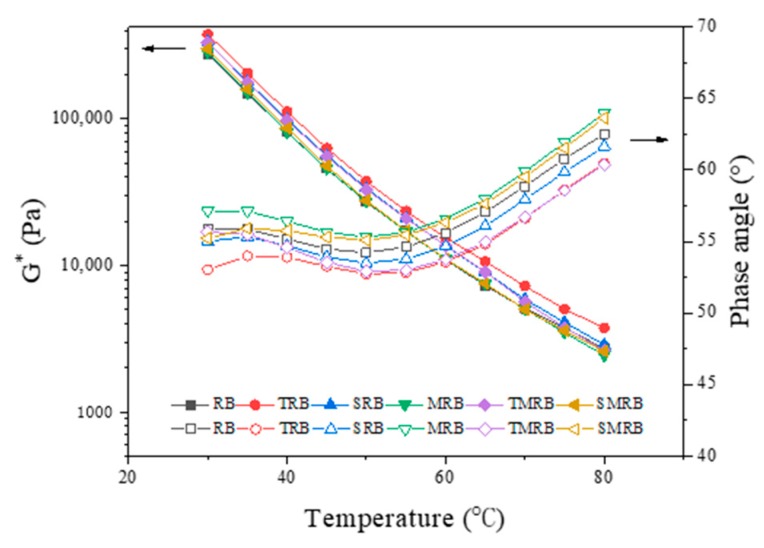
Temperature dependence curve of complex modulus and phase angle of rubberized bitumen.

**Figure 11 materials-12-03538-f011:**
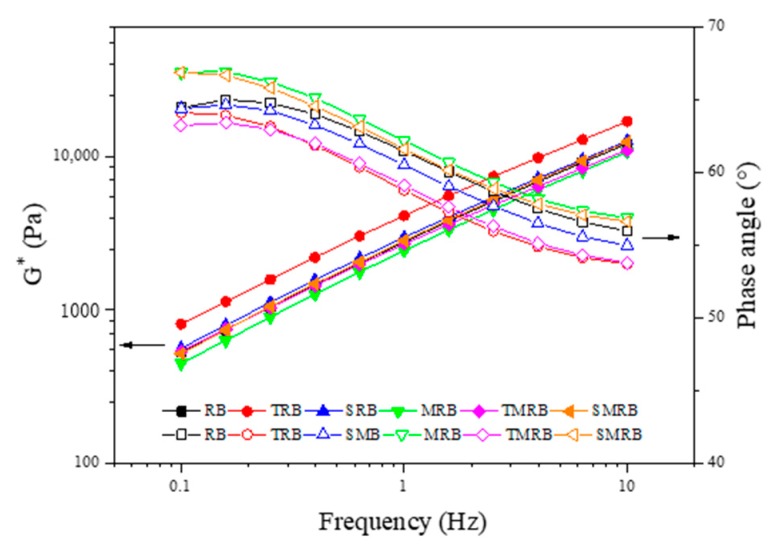
Frequency dependence curve of complex modulus and phase angle of rubberized bitumen.

**Figure 12 materials-12-03538-f012:**
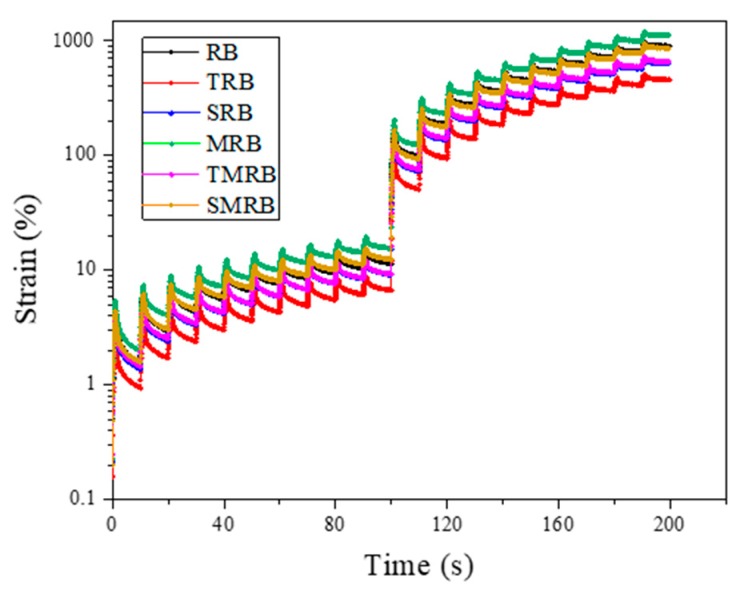
Multiple stress creep recovery (MSCR) test loading results of rubberized bitumen at stress levels of 0.1 and 3.2 kPa.

**Table 1 materials-12-03538-t001:** Technical indices of SK-90 # bitumen.

Test Item		Unit	Results	Test Basis
Penetration (25 °C, 100 g, 5 s)		0.1 mm	94.5	T0604
Ductility (15 °C, 5 cm/min)		cm	>150	T0605
Softening pointT_R&B_		°C	45.9	T0606
Solubility		%	99.6	T0607
Brookfield viscosity (135 °C)		Pa∙s	1.885	T0625
Flash point		°C	260	T0611
Density		g/cm^3^	1.030	T0603
RTFO aging	Mass change	%	+0.4	T0610
	Residual ductility (10 °C)	cm	12	T0605
	Residual penetration ratio (25 °C)	%	57.8	T0604

**Table 2 materials-12-03538-t002:** Technological tests, normative and devices performed on bitumen samples.

Items	Test	Standard	Details	
Physical properties	Softening point	ASTM D36	-	
	Elastic recovery	ASTM D6084	Temperature: 25 °C	
	Force-ductility	-	Temperature: 25 °C	
Rheological properties	Viscosity	ASTM D4402	Spindle number: 24 Specimen weight: 11.5 g	
	Temperature sweep	ASTM D 7175	Plates diameter: 25 mm Gap: 1 mm	Temperature: 30~80 °C Frequency: 0.1 Hz
	Frequency sweep	-		Temperature: 60 °C Frequency: 0.1~10 Hz
	MSCR	ASTM D7405		Temperature: 60 °C

**Table 3 materials-12-03538-t003:** *J*_nr_ and *P* obtained from the MSCR test.

Items	*P* (%)		*J*_nr_ (1/kPa)	
	0.1 kPa	3.2 kPa	0.1 kPa	3.2 kPa
RB	76.9	45.3	0.456	0.509
TRB	79.5	58.1	0.273	0.329
SRB	78.0	52.3	0.336	0.406
MRB	76.7	42.5	0.523	0.603
TMRB	78.2	52.8	0.358	0.430
SMRB	72.5	47.0	0.439	0.511

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
