# Peer review of "A Study on Physical and Rheological Properties of Rubberized Bitumen Modified by Different Methods"

_materials, 2019, doi:10.3390/ma12213538_

Round 1

Reviewer 1 Report

Thank you very much for this interesting paper. The results are overall sound and thouroghly discussed. However, please consider following comments:

for you European readers, the term "asphalt" is usually applied for "asphalt mixtures". Your paper is about the binder. Please consider to change all words "asphalt" to "bitumen", as the paper deals with the binder in detail without any mixture tests. Please check the font size in tables, figure captions as well as in conclusions. It seems the font is too large. Page 3, line 1: the word "and" is too much in the sentence? Please refer to the maximum diameter of te crumb rubber particles, as "number 40 mech size" is not applied internationally. In table 1 you refer to test standard. Can you give a reference to these? Please describe the devices used for stirring and shearing of the modified binder in more detail. Is it a usual stirrer or a dispersing tool? Please give some comment to the DSR configuration (gap size 1 mm in reference to the CR particle size, which seems of similar size. Your present single test result for each test and sample. Are these mean values or base on only single tests. If it is mean value, you could add the information of the number of single values included in tables 1 and 2 and also give some indication on the test result scatter or the test precision. This will help to estimate if the differences in th test results are rather small or significant. (as you did for the force ductility tests) You conclude some effects on low temperature cracking resistance, whereas the lest test temperature applied is 5 °C. Considering the base bitumen with a penetration of ~100, there should be no problem with low-temperature cracking at 5°C. Also the sentece, that the base binder is brittle at 5 °C (page 9, line 198) seems strange. I would recommend to delete all discussion and conclusion about low-temperature cracking resistance, as this property is not analysed within the experiments.

Reviewer 2 Report

The manuscript reports experimental results of asphalt with CR. The results indicate interesting phenomena, but I have some questions as listed below.

1) Are the results in Fig 3 the average values? Is the deviation of each sample smaller than the deviation among the samples?

2) Although it is said "incorporating CR into base asphalt increased its elastic recovery dramatically" (P7, L179), what is the recovery value of the asphalt without CR? Without the information, the reader cannot believe it was a dramatic change.

3) If the recoverability is related to "mechanical properties of CR" (P7, L183), please give the information about the mechanical properties of CRs with and without microwave irradiation. Also, please note what "mechanical properties" exactly mean. Are they the value of Young's modulus? yield stress? Poisson's ratio? or anything else???

4) "The values of G* increased as the temperature increased" (P11, L276) Is it suggested from Fig. 10? For me, it seems G* decreased.

5) Although the authors concluded that "improved low temperature crack resistance" (P13, L326), how can we discuss the low temperature crack resistance? From the words "elastic recovery can reduce low temperature cracking [37]"(P7, L178), we can understand from Fig. 4 that there was not much difference in crack resistance, but TRA is a little better. Therefore, before the conclusion, please show the evidence more clearly.

6) Although the difference of the properties among TOR, Sasobit and microwave, the mechanisms that make the difference are rarely discussed. Only the surface of the CR as shown in Fig. 2 is important? Then, what happens to the surface after high temperature heating in the sample preparation process? Does surface roughness remain even after heating? I think 200-degree centigrade is high enough to change the surface morphology. The authors should discuss what types of chemistry or physics generate the difference TRA, SRA and MRA, or at least, give some suggestions.

Reviewer 3 Report

Review: infrastructures-537814-peer-review-v1

Principal Component Neural Networks for Modeling, Prediction, and Optimization of Hot Mix Asphalt Dynamics Modulus

The authors investigated impact of different modify methods on physical and rheological properties of rubberized asphalt. Six rubberized asphalt samples were prepared including RA, TRA, SRA, MRA, SMRA and TMRA. The softening point test, elastic recovery test and force-ductility test (FDT) was conducted on the prepared samples.

Despite minor comments described below, the paper is very well written; however, the proofreading and improving the language of the paper is inevitable. I think the authors work is interesting.

Comments:

Line 22: in the abstract, dash (-) can be removed. Line 32 and 34: Insert space before the citation number. Please correct this throughout the paper. Line 377: Briefly add one or two sentences about what wet and dry processes are. Line 69: Introduce every acronym before using it in the text. The first time you use the term, put the acronym in parentheses after the full term.\ Line 74: "led the asphalt pavement cracking" is grammatically wrong. Please correct it. Please proofread the whole paper. Line 93: Tab. 1. Use the same symbol as the table caption. Line 112 (Table 1.) Please use the same font size of the text body in the table. Line 174 Figure y-axis label typo: "retio" --> ratio Line 174: Please remove all white lines before figure captions. Fig 5. D4 index. Line 211: space between number and unit symbol. Please check all paper. 22.5mm -->22.5 mm Lines 260, 261,263: check Centigrade symbol. Line 282: The font of Figure Captions is large compare to the text body. Please make it at most the same size of text body. Also, for axis labels, legends and any text in figures. Line 426: Correct the reference. Lines 16, 18, and 41: Please check authors’ names. Line 297: Keep the space between Fig. Tab. and numbers. Fig.12 --> Fig. 12 ( both in caption and text) Line 300: correct subscript nr in "jnr". Line 317: Please keep the same font size throughout the paper. I believe the authors can add a few more recent references on the topic including: a. Pouranian, M. Reza, and Mehdi Shishehbor. "Sustainability assessment of green asphalt mixtures: a review." Environments 6.6 (2019): 73.  b. Pouranian, M. Reza, Reza Imaninasab, and Mehdi Shishehbor. "The effect of temperature and stress level on the rutting performance of modified stone matrix asphalt." Road Materials and Pavement Design (2018): 1-13.

To conclude, I find that, apart from the issues I’ve elaborated above, the paper is fit for publication after addressing the issues.

Reviewer 4 Report

These additives have been used and studied thoroughly in previous studies in this field, what is the actual contribution of the manuscript to the art? I think the discussion on the literature section should clearly indicate how this work differentiates from previous efforts and what is the work adding to the art.

To carry out this comparative study, the authors should have designed a more systematic experiment to compare to previously reported results. What was the ultimate goal of the study, to produce more results or to gain insights into which additive is actually optimum for a given property enhancement. I don’t see the clear objective on the manuscript, or in other words how does the conclusions and observations described in the manuscript fit the objective. If one were to ask which of these approaches is more effective and economical what would be the authors answer? Which of the additives provides better storage stability at high temperature and which one provides higher viscosity. The objectives and conclusions are not aligned.

The authors describe a 40-mesh size particles, however the results do not show anything about the particle size distribution of CR used, did the authors attempt to characterize the particles?

Some statements are made in the manuscript regarding the effect of microwave irradiation, however there are no analysis results shown to prove that the particles actually change chemically as described, only references to the literature and a few SEM images to describe physical, surface changes of the CR particles.

Round 2

Reviewer 4 Report

Questions were addressed.

Minor typo on page 4 line 106, word mothed does not make sense.